# Comparison of Multidrug Use in the General Population and among Persons with Diabetes in Denmark for Drugs Having Pharmacogenomics (PGx) Based Dosing Guidelines

**DOI:** 10.3390/ph14090899

**Published:** 2021-09-03

**Authors:** Niels Westergaard, Lise Tarnow, Charlotte Vermehren

**Affiliations:** 1Centre for Engineering and Science, Department of Biomedical Laboratory Science, University College Absalon, Parkvej 190, 4700 Naestved, Denmark; 2Steno Diabetes Center, Birkevaenget 3, 3rd, 4300 Holbaek, Denmark; litar@regionsjaelland.dk; 3Department of Clinical Pharmacology, University Hospital of Copenhagen, Bispebjergbakke 23, 2400 Copenhagen, Denmark; charlotte.vermehren@regionh.dk; 4Department of Drug Design and Pharmacology, Faculty of Health and Medical Sciences, University of Copenhagen, Universitetsparken 2, 2100 Copenhagen, Denmark

**Keywords:** pharmacogenomics, polypharmacy, persons with diabetes, drug–drug interactions, drug–gene interactions, cytochrome P450, SLCO1B1, drug interaction checkers

## Abstract

Background: This study measures the use of drugs within the therapeutic areas of antithrombotic agents (B01), the cardiovascular system (C), analgesics (N02), psycholeptics (N05), and psychoanaleptics (N06) among the general population (GP) in comparison to persons with diabetes in Denmark. The study focuses on drugs having pharmacogenomics (PGx) based dosing guidelines for CYP2D6, CYP2C19, and SLCO1B1 to explore the potential of applying PGx-based decision-making into clinical practice taking drug–drug interactions (DDI) and drug–gene interactions (DGI) into account. Methods: This study is cross-sectional, using The Danish Register of Medicinal Product Statistics as the source to retrieve drug consumption data. Results: The prevalence of use in particular for antithrombotic agents (B01) and cardiovascular drugs (C) increases significantly by 4 to 6 times for diabetic users compared to the GP, whereas the increase for analgesics (N02), psycoleptics, and psychoanaleptics (N06) was somewhat less (2–3 times). The five most used PGx drugs, both in the GP and among persons with diabetes, were pantoprazole, simvastatin, atorvastatin, metoprolol, and tramadol. The prevalence of use for persons with diabetes compared to the GP (prevalence ratio) increased by an average factor of 2.9 for all PGx drugs measured. In addition, the prevalence of use of combinations of PGx drugs was 4.6 times higher for persons with diabetes compared to GP. In conclusion, the findings of this study clearly show that a large fraction of persons with diabetes are exposed to drugs or drug combinations for which there exist PGx-based dosing guidelines related to CYP2D6, CYP2C19, and SLCO1B1. This further supports the notion of accessing and accounting for not only DDI but also DGI and phenoconversion in clinical decision-making, with a particular focus on persons with diabetes.

## 1. Introduction

Personalized medicine denotes a paradigm shift within medicine that addresses the patient’s individual situation and, most notably, the genetic predispositions of patients in terms of metabolic differences in, e.g., the cytochrome P450 (CYP450) drug-metabolizing enzymes [1,2], leading to variability in drug response [1,2,3]. Diabetes is a complex, chronic illness requiring continuous medical care with multifactorial risk-reduction strategies beyond glycemic control [4]. The prevalence of diabetes continues to increase in virtually all regions of the world, with more than 415 million people worldwide now living with diabetes [5,6]. In Denmark, it is estimated to be 280.000 people [7]. Elderly people, in particular, are more prone to develop diabetes concomitantly leading to associated multiple chronic conditions such as hypertension, dyslipidaemia, coronary heart disease, depression and chronic kidney disease [6,8,9]. In order to prevent, treat and relieve these conditions the introduction of polypharmacy, including prescription cascades and inappropriate medication [8,9,10], is inevitable and so is the occurrence of adverse drug reactions (ADR) and drug–drug interactions (DDI) [8,9,11,12].

Not surprisingly, polypharmacy has been shown to be a significant precipitating factor in frequent hospital admissions [13] and increased risk of mortality [14]. Initiatives both internationally and in Denmark have been taken to incite the best clinical management of patients with multimorbidity and polypharmacy [15,16,17]. These initiatives are, however, often complicated by requiring multiple specialists to be involved in care planning and execution [18]. Therefore, any action that can improve the medical treatment of polypharmacy patients should be carefully considered as a valuable tool to obtain appropriate drug treatment.

CYP450 drug metabolising enzymes are responsible for catalysing the oxidative biotransformation of a large fraction of drugs in daily clinical use to either inactive metabolites or active substances from pro-drugs [19]. In particular, CYP2D6 and CYP2C19 have attracted considerable attention as the major targets for pharmacogenomics (PGx)-based testing because they are highly polymorphic and have been shown to affect both drug response and ADR [2,3,20]. The pharmacogenetic impact on the interaction between drug and CYP450 isozymes, referred to as drug–gene interaction (DGI), has been incorporated into clinical actionable dosing guidelines (AG) and non-actionable dosing guidelines (N-AG) for specific DGIs (see PharmGKB) [21]. Accordingly, a person can be scored as “poor metaboliser” (PM), “intermediate metaboliser” (IM), “extensive metaboliser” (EM; normal activity) and “rapid or ultra-rapid metaboliser” (RM and UM) with UM having faster metabolic activity than RM [22,23,24]. In addition, single nucleotide polymorphisms (SNP) in the solute carrier organic anion transporter 1B1 (*SLCO1B1*) correlate with an increase in the plasma exposure to statins which can lead to muscle toxicity, a common statin-related ADR occurring in 1–5% of exposed users [25] in a dose-dependent fashion. Since statins are some of the most commonly prescribed drugs [25], many people are potentially affected by muscle-related ADR. PGx-based AGs are available for the phenotypes having an intermediate or low function of SLCO1B1 [25]. Daily exposure of patients to drugs having AG is not at all negligible as shown previously [26,27,28,29,30] and additionally makes a significant contribution to the occurrence of side effects [28,29]. In particular, the elderly part of the population is exposed to drugs or drug combinations for which there exist AGs related to PGx of CYP2D6 and CYP2C19 and SLCO1B1 [29,30]. Recently, we have demonstrated that the use of clopidogrel and proton pump inhibitors (PPIs), both having PGx-based AG and FDA annotations, either given alone or in combination is quite widespread, in particular among persons with diabetes and the elderly in Denmark [31]. The aim of this study is to further measure and scrutinize the use of drugs within the therapeutic areas of antithrombotic agents (B01), the cardiovascular system (C), analgesics (N02), psycholeptics (N05) and psycoanaleptics (N06) among the general population in comparison to persons with diabetes in Denmark and with a particular focus on of drugs having PGx-based dosing guidelines to further explore the potential of applying PGx-based decision-making into clinical practice.

## 2. Results

According to the ATC nomenclature, A10 denotes “drugs used in diabetes” which can be subdivided into A10A “insulins and analogues” and A10B “blood glucose lowering drugs excl. insulins”. In this study, persons with diabetes are identified by looking at individuals who redeemed drug prescriptions of A10 during 2018 at a Danish pharmacy. Altogether, 258,494 persons were identified out of a total Danish population of 5,781,190 inhabitants. This corresponds to 4.5% of the Danish population. Table 1 shows the age distribution, as well as the total consumption of A10, A10A, A10B and A10A/B (persons who have redeemed both A10A and A10B), expressed as the number of users and prevalence of use (diabetic users/1000 inhabitants). The number of users is additive horizontally, so the total number of users of A10 is the sum of users of A10A, A10B, and A10A/B. The table illustrates how the number of users and the prevalence of use increase with age—in particular, for users of A10B. This group, as well as A10A/B, have a significant onset in drug use in the age group of 45–64 years. Relative to A10, 16.2% of the users with diabetes redeemed drug prescriptions of A10A, 67.5% of A10B and 16.3% the combination of A10A/B.

Table 2 and Table 3 show the use and prevalence of use of different pharmacological drug classes measured at different levels of ATC codes covering antithrombotic agent’s (B01), the cardiovascular system (C), analgesics (N02), psycholeptics (N05) and psychoanaleptics (N06) both in the general population and among persons with diabetes. It is especially within these ATC groups that PGx-based AGs and N-AGs occur for CYP2D6, CYP2C19 and SLCO1B1. Examples of specific drugs (ATC level 5) having AGs representing each drug class are also given. The prevalence of use shown in Table 2 and Table 3 is expressed relative to the total number of users of A10, A10A, A10B and A10A/B, respectively, as displayed at the bottom of Table 1. The prevalence of use in particular for antithrombotic agents (B01) and cardiovascular drugs (C) increases significantly by 4 to 6 times for users of A10 compared to the general population (Table 2), whereas the increase for analgesics (N02), psycholeptics and psychoanaleptics (N06) was somewhat less: 2–3 times, but still significant (Table 3). Comparison of users of A10A with users of A10B showed that the prevalence of use of the combinations of the different drug classes was mostly higher for users of A10B as shown in Table 2 and Table 3, except for clopidogrel (same) and lower for antihypertensives, opioids, oxycodone, gabapentin, and amitriptyline. A similar comparison of users of A10B with users of A10A/B showed that the prevalence of use was higher and more pronounced for all drug combinations for users of A10A/B both when compared to users of A10A and A10B.

Table 4 shows the use and prevalence of use of the most frequently prescribed PGx drugs having AGs or N-AGs for CYP2D6, CYP2C19 and SLCO1B1 in the general population and among persons with diabetes (A10) sorted by ATC codes. The five most used drugs both in the general population and among persons with diabetes were pantoprazole, simvastatin, atorvastatin, metoprolol, and tramadol, however, the order was different between the two groups. The prevalence of use for persons with diabetes compared to the general population (prevalence ratio) increased by an average factor of 2.9 for all drugs ranging from 1.7 for sertraline to as high as 6.2 for simvastatin except for methylphenidate and atomoxetine. Note that the number of users for the different drugs shown in the table is not additive (vertically) since dispensing to the same users can occur for the different drugs.

Figure 1 shows the use of sertraline, having PGx-based AG for CYP2C19, and tramadol, having AG for CYP2D6, respectively, redeemed either alone or in combination, expressed as the total number of users and prevalence (numbers in brackets) in the general population and among persons with diabetes (A10). As can be seen, the prevalence of use of the combination of sertraline and tramadol was three times higher for persons with diabetes compared to the general population. When the prevalence of use of the combination of sertraline and tramadol was expressed relative to sertraline, 8.8% of the users of sertraline also obtain tramadol, whereas, when expressed relative to tramadol, it was less (4.6%). The same numbers for persons with diabetes were 15.4% and 5.0%, respectively. By calculating the relative risk (RR), it can be seen that persons with diabetes using sertraline have a 1.74 times higher risk of obtaining it in combination with tramadol compared to the general population whereas the same number for diabetic tramadol users is lower but still significant.

Table 5 is based on principles outlined in Figure 1, showing the prevalence of use of drug combinations of the most frequently redeemed drugs in each ATC category except for ondansetron and amiodarone (see Table 4) among the general population and among persons with diabetes. From the table, it can be calculated, based on the principles outlined above, that the prevalence of use of all combinations shown are on average 4.6 times higher for persons with diabetes compared to the general population, whereas the same number, when drugs are given alone (left column), is on average 3.3 higher. The lowest value was 1.6 for the combination of sertraline and quetiapine and the highest was 7.1 for the combination of simvastatin and quetiapine. Importantly, the RR of obtaining a combination of drugs was significantly higher for the majority of the combinations shown in the table for persons with diabetes compared to the general population. For, e.g., diabetic users of sertraline, the RR of obtaining it in combination with clopidogrel, metoprolol, or simvastatin was 2.35, 2.56, and 3.65, whereas for users of clopidogrel, metoprolol or simvastatin the RRs of obtaining these drugs in combination with sertraline were 1.06, 1.05, and 1.01. In addition, by using the drug interaction tracker by Medscape^®^ [32], several of the combinations shown (in bold) are scored as “monitor close”.

## 3. Discussion

In previous studies, it has been shown that the Danish Register of Medicinal Product Statistics constitutes a valuable tool to obtain detailed information, not only about the use of prescription drugs but also about the use of combinations, including drugs having PGx based AGs and N-AGs [28,31]. This offers a unique opportunity to measure drug use in specific disease areas such as diabetes. Based on nationwide registers, the number of persons with diabetes in Denmark in 2017 was estimated to be about 280.000, corresponding to 5% of the population, where type 1 diabetes (T1D) constituted about 28.000 (0.5%) and type 2 diabetes (T2D) about 252.000 (4.5%) [7]. In this study, we identified the total number of individual users of A10 drugs during 2018, which is assumed due to the length of the measured period, to represent a surrogate number for the total diabetes population in Denmark who are in medical antidiabetic treatment. With this assumption, and based on the pharmacological approaches and guidelines for the glycemic treatment of diabetes [33,34], users of solely A10A are T1D and users of solely A10B and both A10A/B are T2D. This assumption seems to be in good alignment with the numbers found by Carstensen et al. [7] both in terms of users, prevalence of use and age-specific prevalence [7]. However, our data on A10 users are slightly lower, somewhat higher for T1D and lower for T2D, which is mainly explained by the different approaches and epidemiological considerations used in this study and by Carstensen et al. [7]. Based on the above, we find it suitable throughout the discussion of the findings of this study to subdivide persons with diabetes into T1D (A10A users), T2D taking no insulin (A10B users) and T2D taking insulin (A10A/B.).

Persons with diabetes have increased platelet reactivity [35,36] and are more prone to cardiovascular disease (CVD) [37,38,39], although there are differences in the underlying pathophysiology between T1D and T2D [38]. This is reflected by the finding of 4–6 times higher prevalence of use of drugs within the drug classes of antithrombotic agents (B01) and the cardiovascular system (C) in persons with diabetes as shown in Table 2 compared to the general population. This clearly underscores the importance of these types of drugs in the prevention and treatment of cardiovascular diseases in persons with diabetes [35,36,37,38,39,40]. Interestingly, when looking at the prevalence’s of use between T1D, T2D taking no insulin and T2D taking insulin it seems to be evident that across most of the ATC categories/drug classes shown, the prevalence of use of antithrombotic agents and CVD drugs was in the order of T2D taking insulin > T2D taking no insulin > T1D. In addition, depression, anxiety and neuropathy are common complications of both T1D and T2D. They affect a large fraction of persons with diabetes and are often associated with poor outcomes [40,41,42,43]. As seen for CVD the underlying pathophysiology for these comorbidities is not well understood, however, the pharmacotherapy for these complications have common features such as the use antidepressants (N06A), i.e., tricyclic antidepressants and serotonin-noradrenaline reuptake inhibitors in addition to gabapentin (and pregabalin)—anticonvulsants normally used to treat epilepsy, and opioids [41,43]. Note that in this study, we cannot discriminate between antidepressants used for neuropathy and depression. Although efficacious in the treatment of neuropathic pain, opioids are not considered to be the first choice because of concerns about abuse and addiction. As was the case with the CVD drugs, persons with diabetes have a 2–3 (for gabapentin 4 times) higher prevalence of use of analgesics including opioids, psycholeptics and psychoanaleptics compared to the general population and essentially follow the same order of prevalence of use as seen for CVD; T2D taking insulin > T2D taking no insulin > T1D. Depression and anxiety seem to be unrecognized and untreated in about two-thirds of persons with diabetes [40,41]. This may reflect the perception among clinicians that psychological matters are less important than physiological matters in persons with diabetes [44], which can explain the higher prevalence of use seen in the CVD area compared to the use of analgesics, psycholeptics and psychoanaleptics. Note that the number of users is not additive (vertical reading) for the different drugs shown in the tables since dispensing to the same users can occur for the different drugs.

However, the clinical relevance and justification of preventing and treating cardiovascular diseases, depression, anxiety and neuropathy by the use of multiple drug regimens also introduce the risk of inappropriate medication that may place persons with diabetes at an increased risk of ADR and poor outcomes [5,11]. Diabetes is inevitably associated with polypharmacy, in particular, among the elderly, and thereby increased risk of frequent hospital admissions [13] and increased risk of mortality [14]. Implementing PGx testing into daily clinical practice can provide a valuable tool to offer “appropriate polypharmacy” as previously suggested among others [3,20,45] and which is in alignment with the recent consensus report on precision medicine in diabetes [46]. In spite of supporting evidence and advances in PGx implementation in clinical practice, evidence on the cost-effectiveness of applying PGx-guided antiplatelet in cardiovascular diseases [47] and in polypharmacy have emerged [48] significant barriers still exist. Mainly concerning physicians’ and pharmacists’ awareness and education, but also evidence level, significance and cost-effectiveness are questioned [49].

The use of drugs in Denmark having PGx-based AGs and N-AGs are quite widespread, especially among the elderly, who often are exposed to several drug combinations having AGs, including combinations having warnings, according to drug–drug interaction checkers such as “monitor closely” or “serious use alternate” [28,29]. Stratifying the use of PGx drugs to persons with diabetes (A10 level) further substantiates the common and by on average 2.9 times more prevalent use of PGx drugs in persons with diabetes compared to the general population. In this study, we do not have data on the prevalence of use of PGx drugs as a function of age intervals. However, since we provide data on the age distribution of users of A10, A10A, A10B and A10A/B (Table 1) we assume that it is the elderly who are the most exposed to PGx drugs, further substantiating age as a key driver of polypharmacy [20,50]. Only in two instances, in the case of methylphenidate and atomoxetine, the prevalence of use was lower for persons with diabetes compared to the general population.

We further scrutinized the consumption of the most used PGx drugs in each drug class (see Figure 1 and Table 5) when drugs were redeemed either alone or in combination from a Danish pharmacy. The prevalence’s of the use of PGx drugs in persons with diabetes were on average 3.3 times higher for diabetic users when given alone. Interestingly, when the PGx drugs were given in combinations, the prevalence ratios increased to an average of 4.6 further suggesting that persons with diabetes are much more exposed to PGx drugs than the general population and in particular, for PGx drug combinations, including drug combinations, for which there exist DDI warnings. Similar findings were also seen for the use of clopidogrel and proton pump inhibitors in persons with diabetes [31]. The frequency of DGI as recently reported for CYP2D6, CYP2C19 and SLCO1B1 [45] further implies that a significant proportion of persons with diabetes will have phenotypes for which actions in principle should be taken regarding dose adjustment or avoidance of the given drugs. Taking phenoconversion into consideration as well, i.e., the combination of DDI and DGI could potentially lead to additional changes in pharmacological responses as has been suggested elsewhere [3]. The differences in RR seen for diabetic users of, e.g., to obtain sertraline in combination with clopidogrel is twice as high as compared to users of obtaining clopidogrel in combination with sertraline, a pattern seen for several of the combinations shown in Table 5 and Figure 1. This suggests that users of certain drugs have a higher probability of obtaining it in combinations with certain other drugs and not necessarily vice versa. The fact that persons with diabetes are more exposed to PGx drugs, both when given alone and in combination, further substantiates that both DGI and DDI, so-called drug–drug-gene interactions (DDGI), are important measures to consider as previously suggested [28,29,31]. This calls for the need for the alignment of drug interaction trackers with regards to the incorporation of DGI and DDGI and thereby considering potential phenoconversion.

A limitation of this register study is a lack of information about dose, compliance, clinical effects as well as the duration of treatments and detailed demographics all of which should be taken into consideration in future research. For data on drug combinations, it cannot be assumed that all users are taking the drugs concomitantly, however, we have supporting data showing that around 50% of drug combinations were redeemed on the same day (unpublished results and [28,31]).

## 4. Materials and Methods

### 4.1. Register Data

This study is a cross-sectional study using The Danish Register of Medicinal Product Statistics [51], which comprises records of all prescriptions redeemed since 1st of January 1996, as the source. Drug consumption data was retrieved with the support of Statistics Denmark [52] for 2018. It is mandatory to report the sale of medicines, and therefore, the data cover all sales in Denmark. The personal identification number [53] (the CPR number) is a unique identifier to all Danish inhabitants which makes it possible to measure a person’s drug consumption. Consumption is expressed as the number of users who redeemed prescriptions of drugs investigated by applying their ATC codes [54]. The drug use among persons with diabetes was identified by measuring inhabitants who redeemed prescriptions of the ATC code A10 (level 2) which solely includes “drugs used in diabetes” including users of A10A (level 3; insulins and analogues) and A10B (level 3; blood glucose-lowering drugs excl. insulins). In addition, the number of users of A10A, A10B and users of both A10A and A10B, referred to as A10A/B, were also measured. By combining the use of A10, A10A, A10B and A10A/B to ATC codes for the drug/drug classes investigated within the therapeutic areas of antithrombotic agents (B01), the cardiovascular system (C), analgesics (N02), psycholeptics (N05) and psychoanaleptics (N06) the number of persons using A10, A10A, A10B and A10A/B alone or in combination with the above-mentioned drug classes were identified and compared to the use in the general population. To convert the number of users to prevalence (users/1000 inhabitants), the total Danish population in 2018 was 5.781.190 and the age group distribution was as follow: 0–17 years 1,165,000; 18–24 years 532,622; 25–44 years 1,441,697; 45–64 years 1,525,308; 65–79 years 859,369 and 80+ years 256,694. The total number of persons who redeemed prescriptions of ATC-code A10 (persons with diabetes) was 258,494 (see Table 1).

Drug–drug interactions were scored in severity by using Medscape^®^ drug interaction checker [32]. Warnings are displayed as “monitor closely” or “serious use alternate”.

The dosing information, length of treatment and indication for prescribing were not recorded, and ethics approval was not applicable according to Danish law since the use of anonymized healthcare data for pharmacoepidemiological research does not require subject consent or approval from Ethics Committee.

### 4.2. Statistics

The relative risk (RR) was calculated by using the MedCalc Software Ltd. relative risk calculator. https://www.medcalc.org/calc/relative_risk.php (Version 20.0.5; accessed on 2 June 2021). The Chi-squared test was performed by using the CHI2.TEST function in Microsoft Excel version 2016.

### 4.3. Clinical Dosing Guidelines

The Clinical Pharmacogenetics Implementation Consortium (CPIC) and the Dutch Pharmacogenetics Working Group (DPWG) clinical dosing guidelines for specific gene-drug pairs were used as the source. The guidelines are available through the publicly available PharmGKB homepage (https://www.pharmgkb.org/ accessed on 15 August 2021). Drugs with guidelines were divided into drugs having an actionable guideline (AG) defined as at least one clinical recommendation (i.e., dose adjustment, dose monitoring or avoidance of the given drug) different from “extensive metaboliser” EM (normal situation) of any of the phenotypes PM, IM or RM. Drugs having a non-actionable guideline (N-AG) were defined as drugs with no clinical recommendation different from EM of any of the phenotypes based on current clinical knowledge.

## 5. Conclusions

The findings of this exploratory cross-sectional register study clearly show that a large fraction of the Danish population and in particular persons with diabetes, especially the elderly, are exposed to drugs or drug combinations for which there exists dosing guidelines as well as FDA annotation related to PGx of CYP2D6, CYP2C19 and SLCO1B1. In addition, it should be emphasized that T2D taking insulin seems to have a higher rate of use of drugs including PGx drugs compared to T2D taking no insulin and T1D. This further supports the notion of the emerging results of accessing and accounting for not only DDI but also DGI, DDGI and phenoconversion as supportive tools in clinical decision-making and appropriate polypharmacy. The focus should be on the elderly, nursing home residents and persons with diabetes due to their high exposure to PGx drugs. In spite of supporting evidence and advances in PGx implementation in clinical practice, including evidence on cost-effectiveness, significant barriers in the Danish healthcare system in implementing the use of PGx, mainly concerning awareness and education, but also at the evidence level which suggests initiatives should be taken focusing on these key barriers.

## Figures and Tables

**Figure 1 pharmaceuticals-14-00899-f001:**
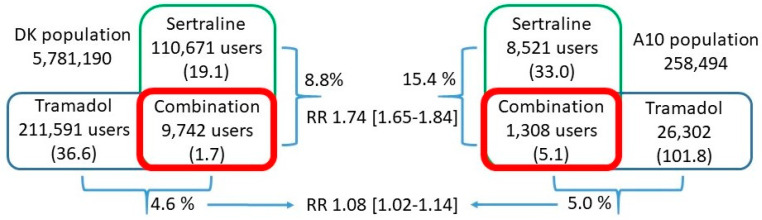
Prevalences of use and relative risks. Note: This figure illustrates the use of sertraline and tramadol either alone or in combination in the general population and among persons with diabetes. Numbers in brackets are prevalence (number of users/1000). Numbers in square brackets are relative risks (RR).

**Table 1 pharmaceuticals-14-00899-t001:** Consumption of drugs used in diabetes.

Age Group	A10	A10A	A10B	A10A/B
0–17	3107 (2.7)	2987 (2.6)	105 (0.1)	15 (<0.1)
18–24	3695 (6.9)	2646 (5.0)	952 (1.8)	97 (0.2)
25–44	23,685 (16.4)	8311 (5.8)	13,153 (9.1)	2221 (1.5)
45–64	94,880 (62.2)	13,194 (8.7)	65,928 (43.2)	15,758 (10.3)
65–79	103,926 (120.9)	10,327 (12.0)	74,102 (86.2)	19,497 (22.7)
80+	29,201 (113.8)	4447 (17.3)	20,262 (78.9)	4492 (17.5)
All	258,494 (44.7)	41,912 (7.3)	174,502 (30.2)	42,080 (7.3)

Note: Data are presented as the total number of users who redeemed prescriptions of the ATC codes A10 (level 2) denoted as “drugs used in diabetes”, A10A (insulins and analogues), A10B (blood glucose-lowering drugs excl. insulins) or the combination thereof 10A/B during 2018. The numbers in brackets show prevalence of use (number of users/1000 inhabitants).

**Table 2 pharmaceuticals-14-00899-t002:** Number of users and prevalence of platelet aggregation inhibitors and cardiovascular drugs.

	Denmark	A10	A10A	A10B	A10A/B
B01 (antithrombotic agents)	556,095 (96.2)	109,300 (422.8)	13,832 * (330.0)	71,648 ^ (410.6)	23,820 (566.0)
B01AC (platelet aggregation inhibitors)	395,373 (68.4)	84,862 (328.3)	10,994 * (261.1)	54,813 ^ (314.1)	19,105 (454.0)
B01AC04 Clopidogrel	127,480 (22.05)	21,746 (84.1)	3363 (80.2)	13,912 ^ (79.7)	4471 (106.3)
C (cardiovascular system)	1,413,160 (244.4)	221,472 (856.8)	26,665 * (636.1)	154,999 ^ (888.3)	39,808 (946.0)
C01 (cardiac therapy)	109,730 (19.0)	22,091 (85.5)	2760 * (65.9)	14,220 ^ (81.5)	5111 (121.5)
C02 (antihypertensives)	17,305 (3.0)	5151 (20.0)	1031 * (24.6)	2785 ^ (16.0)	1385 (31.7)
C03 (diuretics)	424,584 (73.4)	80,925 (313.1)	11,316 * (270.0)	52,129 ^ (298.7)	17,480 (415.4)
C07 (beta blocking agents)	385.920 (66.8)	71.406 (276.3)	7981 * (190.4)	48,563 ^ (278.3)	14,862 (353.2)
C07AB02 (Metoprolol)	279,767 (48.4)	52,559 (203.3)	5783 * (138.0)	35,906 ^ (205.8)	10,870 (258.3)
C08 (calcium channel blockers)	427,655 (74.0)	78,955 (305.4)	9551 * (227.8)	53,536 ^ (306.8)	15,868 (377.1)
C09 (agents acting on the renin-angiotensin system)	747,141 (129.2)	157,696 (610.1)	17,751 * (423.5)	108,958 ^ (624.4)	30,987 (736.4)
C10 (lipid modifying agents)	663,711 (114.8)	174,753 (676.0)	18,752 * (447.4)	122,359 ^ (701.2)	33,642 (799.5)
C10AA (statins)	649,020 (112.3)	171,188 (662.3)	18,039 * (430.4)	120,341 ^ (689.7)	32,808 (779.7)
C10AA01 (Simvastatin)	309,936 (53.6)	86,531 (334.8)	9106 * (217.3)	60,696 ^ (347.8)	16,729 (397.6)
C10AA05 (Atorvastatin)	304,764 (52.7)	76,599 (296.39)	7791 * (185.9)	54,606 ^ (312.9)	14,202 (337.5)

Note: Data are presented as the total number of users who redeemed drug prescriptions of the ATC codes A10, A10A, A10B and A10A/B in combination with antithrombotic agents (B01) and cardiovascular drugs (C). Numbers in brackets are prevalence (number of users/1000). * *p* < 0.05; A10A different from A10B; ^ *p* < 0.05 A10B different from A10A/B when compared horizontally (chi-square test).

**Table 3 pharmaceuticals-14-00899-t003:** Number of users and prevalence of analgesics, psycholeptics, and psychoanaleptics.

	Denmark	A10	A10A	A10B	A10A/B
N02 (analgesics)	1,236,170 (213.8)	124,260 (480.7)	16,453 * (392,6)	83,676 ^ (479.5)	24,131 (573.5)
N02A (opiods)	390,614 (67.6)	47,006 (181.9)	7666 * (182.9)	29,130 ^ (166.9)	10,210 (242.6)
N02AA05 (Oxycodone)	79,328 (13.7)	9536 (36.9)	1856 * (44.3)	5469 ^ (31.3)	2211 (52.5)
N02AX02 (Tramadol)	211,591 (36.6)	26,302 (101.8)	3809 * (90.9)	16,697 ^ (95.7)	5796 (137.7)
R05DA05 (Codeine)	84,210 (14.6)	8987 (34.8)	1156 * (27.6)	6091 ^ (34.9)	1740 (41.4)
N02B (other analgesics and antipyretics)	1,089,807 (188.5)	113,995 (441.0)	14,711 * (351.0)	76,963 ^ (441.0)	22,321 (530.4)
N03AX12 (Gabapentin)	78.048 (13.5)	11.559 (44.9)	1.958 * (46.7)	6.640 ^ (38.1)	3.001 (71.3)
N05 (psycoleptics)	407,387 (70.5)	37,461 (144.9)	5550 * (132.4)	25,042 ^ (143.5)	6869 (163.2)
N05A (antipsychotics)	131,836 (22.8)	13,355 (51.7)	1877 * (44.8)	8903 ^ (51.0)	2575 (61.2)
N05B (anxiolytics)	124,731 (21.6)	11,906 (46.1)	1802 * (42.9)	8079 (46.3)	2025 (48.2)
N05C (hypnotics and sedatives)	232,933 (40.3)	21,058 (81.5)	3407 * (81.3)	13,618 ^ (78,0)	4033 ^ (95.8)
N06 (psychoanaleptics)	471,341 (81.5)	44,440 (171.9)	6699 * (159.8)	28,961 ^ (166.0)	8780 (208.7)
N06A (antidepressants)	416,064 (72.0)	41,942 (162.3)	6188 * (147.6)	27,388 ^ (157.0)	8366 (198.8)
N06AA09 (Amitriptyline)	34,598 (6.0)	4334 (16.8)	693 * (16.5)	2555 ^ (14.6)	1086 (25.8)
N06AX21 (Duloxetin)	34,277 (5.9)	3852 (14.9)	533 * (12.7)	2514 ^ (14.4)	805 (19.1)

Note: Data are presented as the total number of users who redeemed drug prescriptions of the ATC codes A10, A10A, A10B and A10A/B in combination with analgesics (N02); gabapentin, psycholeptics (N05) and psychoanaleptics (N06). Numbers in brackets are prevalence (number of users/1000). * *p* < 0.05; A10A different from A10B; ^ *p* < 0.05 A10B different from A10A/B when compared horizontally (chi-square test).

**Table 4 pharmaceuticals-14-00899-t004:** Consumption of PGx drugs in the general population (GP) and among persons with diabetes (A10).

Drug Name	PGx-G	ATC	Users (GP)	Prevalence (GP)	Users (A10)	Prevalence (A10)	Prevalence Ratio
Pantoprazol	AG	A02BC02	329,222	56.95	39,287	151.98	2.7
Lansoprazol	AG	A02BC03	135,980	23.52	17,246	66.72	2.8
Omeprazol	AG	A02BC01	119,274	20.63	14,286	55.27	2.7
Esomeprazol	N-AG	A02BC05	32,295	5.59	3054	11.81	2.1
Ondansetron	AG	A04AA01	13,979	2.42	1341	5.19	2.2
Clopidogrel	AG	B01AC04	127,480	22.05	21,746	84.13	3.8
Amiodaron	N-AG	C01BD01	8582	1.48	1420	5.49	3.7
Metoprolol	AG	C07AB02	279,767	48.39	52,559	203.33	4.2
Carvedilol	N-AG	C07AG02	33,506	5.80	8004	30.96	5.3
Bisoprolol	N-AG	C07AB07	24,953	4.32	4860	18.80	4.4
Atenolol	N-AG	C07AB03	15,517	2.68	2859	11.06	4.1
Simvastatin	AG	C10AA01	309,936	53.61	86,531	334.75	6.2
Atorvastatin	AG	C10AA05	304,764	52.72	76,599	296.33	5.6
Tramadol	AG	N02AX02	211,591	36.60	26,302	101.75	2.8
Codein	AG	R05DA04	84,210	14.57	8987	34.77	2.4
Oxycodon	N-AG	N02AA05	79,328	13.72	9536	36.89	2.7
Quetiapine	N-AG	N05AH04	65,208	11.28	5540	21.43	1.9
Olanzapine	N-AG	N05AH03	17,584	3.04	1819	7.04	2.3
Risperidon	N-AG	N05AX08	16,066	2.78	1881	7.28	2.6
Aripiprazol	AG	N05AX12	12,381	2.14	1347	5.21	2.4
Sertraline	AG	N06AB06	110,671	19.14	8521	32.96	1.7
Citalopram	AG	N06AB04	90,460	15.65	9824	38.00	2.4
Mirtazapin	N-AG	N06AX11	83,603	14.46	9035	34.95	2.4
Venlafaxin	AG	N06AX16	48,398	8.37	5307	20.53	2. 5
Methylphenidate	N-AG	N06BA04	38,620	6.68	984	3.81	0.6
Amitriptyline	AG	N06AA09	34,598	5.98	4334	16.77	2.8
Duloxetine	N-AG	N06AX21	34,277	5.93	3852	14.90	2.5
Escitalopram	AG	N06AB10	23,607	4.08	2153	8.33	2.0
Nortriptyline	AG	N06AA10	14,339	2.48	1718	6.65	2.7
Paroxetine	AG	N06AB05	12,410	2.15	1332	5.15	2.4
Fluoxetine	N-AG	N06AB03	10,535	1.82	831	3.21	1.8
Atomoxetine	AG	N06BA09	9778	1.69	212	0.82	0.5

Note: Only drugs redeemed by more than 8000 users in the general population are shown and compared to persons with diabetes. Drugs are sorted by ATC categories. AG; actionable dosing guideline, N-AG; non-actionable dosing guideline, GP; general population.

**Table 5 pharmaceuticals-14-00899-t005:** Prevalences of use and relative risks (RR).

Drug Name	Alone	Clopidogrel	Metoprolol	Pantoprazole	Quetiapine	Sertraline	Tramadol	Simvatsatin
Clopidogrel	84.1/22.1 (3.8)		25.7/4.7 (5.4)	**20.4/4.4 (4.6)**	2.1/0.5 (4.1)	**4.0/1.0 (4.0)**	12.1/2.6 (4.7)	31.5/7.1 (4.4)
RR			1.42 [1.39–1.45]	1.21 [1.18–1.24]	1.09 [0.99–1.19]	1.06 [1.00–1.13]	1.22 [1.18–1.26]	1.17 [1.14–1.19]
Metoprolol	203.3/48.4 (4.2)	25.7/4.7 (5.4)		41.7/8.4 (4.9)	3.6/0.8 (4.7)	**6.8/1.5 (4.4)**	26.7/5.2 (5.2)	76.2/11.6 (6.6)
RR		1.29 [1.26–1.32]		1.18 [1.16–1.20]	1.11 [1.04–1.20]	1.05 [1.00–1.10]	1.23 [1.20–1.26]	1.56 [1.54–1.59]
Pantoprazole	152.0/56.9 (2.7)	**20.4/4.4 (4.6)**	41.7/8.4 (4.9)		5.3/1.7 (3.2)	7.4/2.5 (3.0)	27.6/8.0 (3.4)	49.8/7.6 (6.6)
RR		1.73 [1.68–1.78]	1.85 [1.82–1.89]		1.18 [1.12–1.25]	1.11 [1.06–1.17]	1.29 [1.26–1.32]	2.47 [2.43–2.51]
Quetiapine	21.4/11.3 (1.9)	2.1/0.5 (4.1)	3.6/0.8 (4.7)	5.3/1.7 (3.2)		3.2/1.9 (1.6)	**3.8/1.3 (3.0)**	7.2/1.0/(7.1)
RR		2.18 [2.00–2.38]	2.47 [2.31–2.63]	1.66 [1.58–1.74]		0.86 [0.80–0.91]	1.57 [1.48–1.67]	3.72 [3.56–3.88]
Sertraline	33.0/19.1 (1,7)	**4.0/1.0 (4.0)**	**6.8/1.5 (4.4)**	7.4/2.5 (3.0)	3.2/1.9 (1.6)		**5.1/1.7 (3.0)**	10.9/1.7 (6.3)
RR		2.35 [2.21–2.51]	2.56 [2.45–2.68]	1.72 [1.66–1.80]	0.94 [0.88–1.01]		1.74 [1.65–1.84]	3.65 [3.52–3.78]
Tramadol	101.8/36.6 (2,8)	12.1/2.6 (4.7)	26.7/5.2/(5.2)	27.6/8.0 (3.4)	**3.8/1.3 (3.0)**	**5.1/1.7 (3.0)**		35.5/5.0 (7.0)
RR		1.67 [1.61–1.74]	1.86 [1.82–1.91]	1.24 [1.21–1.27]	1.07 [1.00–1.14]	1.08 [1.02–1.14]		2.53 [2.49–2.58]
Simvastatin	334.8/53.6 (6.2)	31.5/7.1 (4.4)	76.2/11.6 (6.6)	49.8/7.6 (6.6)	7.2/1.0 (7.1)	10.9/1.7 (6.3)	35.5/5.0 (7.0)	
RR		0.71 [0.70–0.73]	1.05 [1.04 1.07]	1.06 [1.04–1.08]	1.13 [1.07–1.19]	1.01 [0.97–1.05]	1.13 [1.10–1.15]	

Note: Data are presented as prevalence (users/1000) in persons with diabetes/in the general population (light blue rows) who redeemed combinations of drugs shown in the upper and left panel. The numbers in brackets are prevalence ratios, i.e., prevalence for the diabetes population divided by prevalence of the general population. The white rows show the relative risk (RR) for persons with diabetes who redeemed the drugs shown in the left panel to be exposed to the combinations of drugs (shown in upper panel). The numbers in the column “alone” are taken from Table 4 for comparison of prevalence’s and prevalence ratio when the drugs are taken alone or in combination. Drug–drug interactions were scored by Medscape^®^ [32] and bold indicates “monitor closely”.

## Data Availability

The data presented in this study are available within the article.

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
