# Peer review of "Comparison of Multidrug Use in the General Population and among Persons with Diabetes in Denmark for Drugs Having Pharmacogenomics (PGx) Based Dosing Guidelines"

_pharmaceuticals, 2021, doi:10.3390/ph14090899_

Round 1

Reviewer 1 Report

Drug use risk assessment is an important health topic. Nowadays patients with multimorbidities are increasing in number bringing new problems some of them related to the use of multiple drugs. The paper analyses the Danish Register of Medicinal Product Statistics and compares drug use in the general population with the diabetic population, focusing on drugs having pharmacogenomics based dosing guidelines. The study brings important findings regarding several drug associations that deserve particular monitoring because of drug-drug interaction and drug-gene interaction. 

It calls to attention the need to conciliate prescriptions and raise awareness of potentially risky drug associations

The paper I've received has the Materials and Methods as chapter 4, (Ln 329...) after Discussion (Ln 210), is this intentional?

Page numbers also have some problems because after page 8, numbering restarts 

Nevertheless, the subject is relevant and the paper is based on a well-conducted study and information obtained is important for a more safe health care

Author Response

Thank you for the positive review.

Page numbering has been corrected.

According to the manuscript template Materials and Methods section appears after the Discussion

Reviewer 2 Report

The authors show a descriptive study without precise associations. It is a study with a limited scope, since it does not manage to raise or expose any new concept. The manuscript presents a limited discussion that does not manage to specifically unite the current concepts in the state of the art.
The data shown may be interesting, but the scope is anedoctic.

Author Response

We don't have any comments to this report

Reviewer 3 Report

The study compare the use of drugs with PGx based dosing recommendation I(focus on CYP2D6, CYP2C19 and SLCOB1) in diabetics with general population. Data from the Danish Register of Medicinal Product Statistics revealed higher prevalence of use of antithrombotic agents and cardiovascular drugs, in addition diabetics are more exposed to combinations of PGx drugs. 

The study findings evidenced that diabetics are at an increased overall risk of DDI or DGI, which may impair pharmacological benefit in this group of patients, suggesting that a PGx-testing may be beneficial in this group of patients.

The research question is relevant, the methodology is adequate, the manuscript is well written. I have no further questions. 

Author Response

Thank you for the positive review

Reviewer 4 Report

Abstract

Please indicate the method of this study. Is this a cross-sectional study?

Methods

This section needs some clarification in terms of the type of research design, the tool used for collecting up data, and measures taken to ensure that all collected data are accurate.

Results

If any demographic data have been collcted, please share in the result's section.

Conclusion

Please separet it from the discussion using a subheading. Please add the implications of your findings for policy-making, practice, and future research. In other words, what should be done based on your findings?

Author Response

Dear reviewer,

Thank you for a positive review and constructive feedback.

In the abstract we have added  info about study design as suggested.

In the method section we have added further clarifications related to your comments on accuracy.

We don't have any additional demographic data besides those shown in Table 1: we have added additional text line 319-321 dealing with this as part of the limitations of this study.

We have inserted the subheading "Conclusion" as suggested and moved our conclusions accordingly and inserted sentences dealing with the implications of our findings and future research directions line 319-321 and 394-398.